# Preparation of Microencapsulated Phase Change Materials from Sulfonated Graphene Stabilized Pickering Emulsion

**DOI:** 10.3390/polym15112441

**Published:** 2023-05-25

**Authors:** Weiping Li, Dajiang Mei, Jihu Wang, Hui Wu, Shaoguo Wen

**Affiliations:** College of Chemistry and Chemical Engineering, Shanghai University of Engineering Science, Shanghai 201620, Chinameidajiang718@pku.edu.cn (D.M.); 15990350982@163.com (H.W.)

**Keywords:** microencapsulated phase change materials, sulfonated graphene, Pickering emulsion

## Abstract

Microencapsulated phase change materials (MCPCM) as a green energy storage material not only prevent leakage of phase change materials but also increase the heat transfer area of phase change materials. Extensive previous work has shown that the performance of MCPCM depends on the shell material and MCPCM with polymers, as the shell material suffers from low mechanical strength and low thermal conductivity. In this study, a novel MCPCM with hybrid shells of melamine-urea-formaldehyde (MUF) and sulfonated graphene (SG) was prepared by in situ polymerization using SG-stabilized Pickering emulsion as a template. The effects of SG content and core/shell ratio on the morphology, thermal properties, leak-proof properties, and mechanical strength of the MCPCM were investigated. The results showed that the incorporation of SG into the shell of MUF effectively improved the contact angles, leak-proof performance, and mechanical strength of the MCPCM. Specifically, the contact angles of MCPCM-3SG were reduced by 26°, the leakage rate was reduced by 80.7%, and the breakage rate after high-speed centrifugation was reduced by 63.6% compared to MCPCM without SG. These findings suggest that the MCPCM with MUF/SG hybrid shells prepared in this study has great potential for application in thermal energy storage and management systems.

## 1. Introduction

Microencapsulated phase change materials (MCPCM) have received considerable attention over the past few decades as a green energy storage material, showing great potential in effectively alleviating and solving energy problems [1]. They not only protect the solid–liquid phase change material from the external environment but also increase its heat transfer area [2,3,4]. To meet diverse application needs, numerous methods have been investigated for preparing MCPCM, including interfacial polymerization [5,6], suspension polymerization [7,8], in situ polymerization [9,10], emulsion polymerization [11,12], etc. Among them, Pickering emulsion polymerization is of particular interest due to its non-additive organic surfactants, environmentally friendly nature, and high emulsion stability [13]. Furthermore, Pickering emulsion polymerization can prepare MCPCM with polymer/inorganic hybrid shells, which have superior mechanical properties and anti-leakage performance compared to microcapsules prepared by conventional methods [14].

On the other hand, a considerable amount of prior research has demonstrated that the efficacy of MCPCM relies on the material used for the shell [15,16,17]. Currently, popular shell materials consist of polymethyl methacrylate (PMMA) [18], melamine-formaldehyde (MF) [19], polystyrene (PS) [20], polyurea (PUA) [21], etc. Among them, melamine-urea-formaldehyde (MUF) resin stands out for its high strength, good heat resistance, and ease of preparation through in situ polymerization. Despite the numerous advantages of MUF as the shell material for MCPCM, it still exhibits limitations inherent to organic shell materials, including poor mechanical strength, low thermal conductivity, and high supercooling [22]. Incorporating inorganic functional particles into the organic shell can effectively address the weaknesses of organic shell materials and synergistically enhance both thermal conductivity and mechanical strength [11]. Sun et al. [23] successfully prepared MCPCM by employing sodium dodecyl sulfate as the emulsifier, n-octadecane as the core material, and MUF and diatomaceous earth as the hybrid shell components. The results of this study show that the incorporation of diatomaceous earth enhances the mechanical strength of MCPCM. At a diatomaceous earth content of 2 wt% relative to the mass of octadecane, the average Young’s modulus of the microencapsulated phase change materials reaches 942.85 MPa, representing a 1.64-fold increase compared to microencapsulated phase change materials without diatomaceous earth. Despite the performance enhancement achieved by introducing functional particles into the organic shell, the preparation of MCPCM still requires the addition of organic surfactants. After the reaction, organic surfactants are often regarded as “impurities” in MCPCM, which limits their application. In contrast, the method of introducing functional particles through Pickering emulsion polymerization demonstrates distinct advantages.

Graphene and its derivatives, which have high intrinsic thermal conductivity [24,25], good mechanical properties, and impermeability, can be utilized to improve the performance of MCPCM when combined with shell materials [26,27,28,29]. The use of graphene as a Pickering stabilizer is severely limited due to its inert surface. When utilizing graphene as a Pickering stabilizer, it is customary to undergo a surface modification of the substance and employ co-emulsifiers to achieve emulsion stability. This process is considered complicated. Microencapsulated phase change materials were successfully prepared by Zhao et al. [11], who employed modified graphene as a Pickering stabilizer and employed styrene-maleic anhydride, polyvinyl alcohol, and polyethylene glycol 800 as co-emulsifiers. The results demonstrated a significant enhancement in the thermal conductivity and leak-proof performance of the microcapsules with the introduction of graphene. Graphene oxide (GO), a notable derivative of graphene, is commonly employed as a Pickering stabilizer in the synthesis of MCPCM. Li et al. [29] employed Pickering suspension polymerization to synthesize thermally stable MCPCM. The MCPCM were composed of n-eicosane as the phase change material, PUA as the shell material, and GO as the stabilizer. The results indicate that the prepared microcapsules exhibit good thermal reliability, with no observed leakage even after 100 cycles of heating and cooling. However, due to the severe disruption of the conjugated network, GO lacks some advanced characteristics of graphene, such as stability [30]. Li et al. [31] employed GO and sulfonated graphene (SG) as Pickering stabilizers to stabilize epoxy resin emulsions. After storage for 60 days, it was observed that the GO-stabilized emulsion exhibited gelation and phase separation, while the SG-stabilized emulsion did not show any significant abnormalities. Currently, there are numerous literature reports on the application of SG-stabilized Pickering emulsions [32,33,34,35]. Nevertheless, few studies have investigated the preparation of MCPCM by Pickering emulsions of SG.

In this study, microcapsules were created containing an n-octadecane (C18) core and a melamine-urea-formaldehyde (MUF) shell via in situ polymerization, utilizing SG, a graphene derivative, as a Pickering emulsion stabilizer. The effects of different SG additions and core/shell ratios on the chemical composition, morphology, thermal properties, mechanical strength, and leak-proof properties of MCPCM were investigated using Fourier transform infrared spectroscopy (FT-IR), polarized light microscopy (POM), scanning electron microscopy (SEM), and differential scanning calorimetry (DSC) characterization methods.

## 2. Materials and Methods

### 2.1. Materials

In this study, n-octadecane (98%) and formaldehyde solution (37–40%) were obtained from Aladdin Reagent Co., Ltd. (Shanghai, China). Analytical grade melamine, urea, and anhydrous citric acid were acquired from Titan Technology Co., Ltd. (Shanghai, China). Sodium dodecyl sulfate (SDS) and triethanolamine, both analytically pure, were purchased from Sinopharm Chemical Reagent Co., (Beijing, China). Suzhou Qualcomm Technology Co., (Suzhou, Jiangsu, China) produced sulfonated graphene (SG) slurry with a concentration of 10 wt%.

### 2.2. Preparation of Pickering Phase Change Material Emulsions

A certain amount of SG slurry was weighed and added to a beaker. Deionized water was then added to the beaker, and the mixture was dispersed using ultrasound for 10 min.

The beaker was then placed in a constant temperature oil bath at 40 °C, and molten C18 was added, followed by mechanical stirring at 10,000 rpm. After 0.5 h of stirring, the resulting emulsion was named PCMEs-1SG, PCMEs-2SG, or PCMEs-3SG, depending on the amount of SG slurry used. The amount of SG slurry in the emulsions PCMEs-1SG, PCMEs-2SG, and PCMEs-3SG was 10 wt%, 20 wt%, and 30 wt% of the amount of C18, respectively. For comparison, an emulsion without SG slurry was prepared using SDS as the emulsifier and was named PCMEs-0SG. The amount of chemicals used is shown in Table 1.

### 2.3. Preparation of Microencapsulated Phase Change Materials

The microencapsulated phase change material (MCPCM) was synthesized as follows:
(1)A predetermined amount of melamine, formaldehyde solution, and 70 mL of deionized water were weighed and combined in a beaker. The pH value of the mixture was adjusted to 8.5 by adding triethanolamine, and the reaction was sustained for 1 h at 70 °C with mechanical stirring. The mixture was cooled to room temperature, and 0.92 g of urea were added to obtain the melamine-urea-formaldehyde (MUF) prepolymer.(2)The phase change material emulsion was then transferred into a three-neck flask and placed in a constant temperature oil bath at 40 °C with mechanical stirring at 350 rpm. The oil bath temperature was gradually elevated to 75 °C, and the MUF prepolymer solution was slowly added dropwise to the three-necked flask for 20–30 min. Upon reaching the predetermined oil bath temperature, a 10 wt% citric acid solution was gradually added dropwise to regulate the pH of the system to approximately 5, and the reaction was carried out for 3.5 h. The pH of the reaction mixture was then adjusted to 9 with triethanolamine to stop the polymerization reaction.


The suspension of MCPCM was then obtained and filtered to obtain the MCPCM. The impurities in MCPCM were removed by washing with deionized water multiple times. Finally, the MCPCM was dried in an oven at 45 °C for 12 h to obtain the MCPCM powder.

The synthesized microcapsules were named MCPCM-0SG, MCPCM-1SG, MCPCM-2SG, MCPCM-3SG, and MCPCM-3SG based on the amount of SG slurry used. Meanwhile, with a fixed amount of SG slurry at 20 wt% of the amount of C18, the synthesized microcapsules were named MCPCM-1/1, MCPCM-2/1, MCPCM-3/1, and MCPCM-4/1 when the core/shell ratio of MCPCM was 1:1, 2:1, 3:1, and 4:1, separately. The amount of chemicals used is shown in Table 2.

In the microcapsule preparation process, chemical reactions occur in two stages: the preparation of prepolymers and the synthesis of polymer shells. The preparation of the prepolymer primarily involves a nucleophilic addition reaction between melamine and formaldehyde under alkaline heating conditions, resulting in the formation of water-soluble hydroxymethyl melamine prepolymer [36]. Additionally, the reaction between urea and formaldehyde leads to the formation of water-soluble hydroxymethyl urea [37]. The corresponding reaction equations are illustrated in Figure 1a,b. The synthesis of the polymer shell is achieved by subjecting hydroxymethyl melamine and hydroxymethyl urea to acidic heating conditions. In this step, the hydroxymethyl group undergoes further condensation reactions with amino (-NH_2_), methylene (-NH-), or hydroxyl groups. These reactions gradually increase the molecular weight and form a cross-linked structure of the water-insoluble MUF polymer [38]. The reaction equation for this step is shown in Figure 1c.

### 2.4. Characterization

The chemical structure of the samples was characterized using Fourier transform infrared spectroscopy (FT-IR, Nicolet iS5, Wilmington, MA, USA). The samples were mixed with KBr, ground, and pressed into tablets to prepare them for analysis.

The distribution and state of the phase change material emulsion at different temperatures were observed using a hot-stage polarizing microscope (PH-PG3230, Hangzhou, Zhejiang, China).

The morphology of the microcapsules was observed using a scanning electron microscope (ZEISS Gemini 300, Oberkochen, Baden-Württemberg, Germany). To prepare the sample, a conductive double-sided adhesive was applied to the sample table, and the MCPCM powder was dispersed in anhydrous ethanol, followed by ultrasonication for 5 min. The resulting mixture was then uniformly coated on a silicon plate. After the ethanol had evaporated, the silicon plate was applied to the double-sided adhesive and vacuum sprayed with gold.

The thermal storage properties and phase transition temperature of MCPCM were analyzed using a differential scanning calorimeter (DSC, STA449F3, Sebl, Wunsiedel, Germany). The test parameters included a temperature range of −10 to 60 °C, a temperature rise and fall rate of 5 °C/min, an N_2_ atmosphere, and Al_2_O_3_ as the reference material. The encapsulation rate (*E_r_*) of MCPCM was determined from the DSC curve and calculated using the following equation [39]:(1)$$E_{r}\left(\%\right)=\frac{{∆H}_{m,MCPCM}+{∆H}_{c,MCPCM}}{{∆H}_{m,C18}+{∆H}_{c,C18}}\times 100\%,$$
where $∆H_{m,C18}$ (J/g) is the enthalpy of melting of n-octadecane, $∆H_{c,C18}$ (J/g) is the enthalpy of solidification of n-octadecane, $∆H_{m,MCPCM}$ (J/g) is the enthalpy of melting of MCPCM, and $∆H_{c,MCPCM}$ (J/g) is the enthalpy of solidification of MCPCM.

The static contact angle of the MCPCM was measured using a contact angle measuring instrument (JCY, Shanghai, China). After being freshly prepared, the MCPCM powder was dried for 12 h at 45 °C and then subjected to contact angle measurements using a contact angle measuring instrument. To prepare the sample for testing, the MCPCM powder was affixed onto a clean glass sheet using double-sided tape, and the contact angle was measured thereafter. The test conditions entailed using a volume of 3 μL of deionized water, with the contact angle being measured at three different positions on each sample. The reported value for each sample was obtained as the average of the three measurements.

The mechanical properties of the MCPCM were tested using a centrifuge (TG16-WS, Changsha, Hunan, China). A specific mass of MCPCM, *m*_0_, was subjected to centrifugation at 8000 rpm for 30 min. After the centrifugation, the samples in the centrifuge tube were washed with ethanol and distilled water and subsequently dried. The breakage rate (*B_r_*) of the MCPCM was then obtained using the following equation [40]:(2)$$B_{r}\left(\%\right)=\frac{m_{0}-m_{1}}{m_{0}}\times 100\%,$$
where $m_{0}$ and $m_{1}$ were the mass of the sample before and after the centrifuge.

The leakage rate (*L_r_*) was used for evaluating the leakage prevention performance of the MCPCM. The experiment was conducted at a temperature of 60 °C in an oven (DHG-9140A, Shanghai, China). The MCPCM were placed on thin square filter papers individually. The MCPCM were taken out of the oven and weighed accurately every 1 h. The leakage rate (*L_r_*) of the MCPCM was then obtained using the following equation [29]:(3)$$L_{r}\left(\%\right)=\frac{M_{0}-M_{t}}{M_{0}}\times 100\%,$$
where $M_{0}$ and $M_{t}$ were the initial mass of the sample and the weight of the sample at time *t*.

## 3. Results and Discussion

### 3.1. Pickering Emulsion Stabilized by SG

The morphology and stability of Pickering emulsions can have an impact on the properties of the resulting MCPCM. The droplet size distribution of the phase change material emulsion was observed to be widely distributed, as shown in Figure 2. As the content of SG increased, the droplet size decreased, indicating that the effectiveness of the Pickering emulsifier is influenced by the SG content.

The morphology of the phase change material emulsion PCMEs-2SG at different temperatures is shown in Figure 3. As shown in Figure 3, the shape of Pickering emulsion droplets varied with temperature. At 10 °C, the droplets appeared as irregular spheres, while at 30 °C and 45 °C, they became almost perfect spheres. It is noteworthy that the shape of the droplets remains undamaged despite changes in temperature, indicating the stability of Pickering emulsions.

### 3.2. Chemical Composition of MCPCM

Figure 4 depicts the FT-IR spectra of C18, MUF, SG, and MCPCM-2SG. C18 displays two strong characteristic peaks at 2853 cm^−1^ and 2923 cm^−1^, which are associated with C-H stretching vibrations. The peak near 1466 cm^−1^ is assigned to the combined effect of asymmetric bending vibrations of -CH_3_ and shear bending vibrations of -CH_2_, while the peak near 720 cm^−1^ corresponds to wobbling vibrations of -CH_2_ [12]. MUF presents a broad and intense absorption peak around 3350 cm^−1^, which is attributed to the superimposition of N-H and O-H bond stretching vibrations. The characteristic peak near 1338 cm^−1^ is due to the C-N stretching vibrations, and the peak at 812 cm^−1^ reflects the characteristic peak of the triazine ring vibrations, which are all typical vibrations of MUF resins functionalities [23]. SG shows a strong and broad absorption peak at 3405 cm^−1^, which corresponds to the stretching vibration of -OH, while the peaks at 1718 cm^−1^ and 1620 cm^−1^ correspond to the stretching vibration of C=O and C=C, respectively [41]. The sulfonic acid group’s characteristic absorption peaks appear at 1180 cm^−1^, 1120 cm^−1^, and 1040 cm^−1^ [42]. MCPCM-2SG exhibits characteristic peaks for C18, MUF, and SG without any new peaks, indicating that no chemical interactions occurred between the core and shell materials.

### 3.3. Morphology of MCPCM

Figure 5 displays the morphology of MCPCM prepared with varying SG contents. The presence of a considerable number of fine particles on the surface of MCPCM-0SG without SG addition, as observed in Figure 5a,e, is attributed to the self-polymerization of some MUF prepolymers [43]. In Figure 5b of MCPCM-1SG, a small number of microcapsules exhibited surface holes, which can be attributed to the instability of the emulsion droplets and the consequent loss of C18 during the prepolymer polymerization process. Proper SG addition leads to the complete coverage of the phase change material by the MCPCM-2SG and MCPCM-3SG shells, with MCPCM-2SG having better sphericity than MCPCM-3SG.

Figure 6 presents the morphology of MCPCM with varying core/shell ratios. As illustrated in Figure 6a, microcapsules with a core/shell ratio of 1:1 exhibited an extremely rough surface with numerous fine particles attached to it. This is likely due to the excess MUF prepolymer resulting from the relatively low core/shell ratio, leading to self-polymerization on the surface of the microcapsules or in the aqueous solution. The core/shell ratio between 2:1 and 3:1, as seen in Figure 6b,c, provides complete microcapsule encapsulation, reducing surface roughness compared to a core/shell ratio of 1:1. However, at a core/shell ratio of 4:1, cracks and pores appear on the surface of some microcapsules, and the microcapsule encapsulation is less effective. The increase in core/shell ratio causes excess C18 emulsion droplets that are not completely encapsulated by the MUF, leading to defective microcapsules with surface cracks.

### 3.4. Thermophysical Properties of MCPCM

The thermal properties of MCPCM were characterized by DSC. Figure 7 shows the DSC curves of C18 and MCPCM. As evident from Figure 7b, the exothermic peak width of MCPCM is significantly greater than that of C18, accompanied by multiple exothermic peaks. This behavior can be attributed to the change in the crystallization mode of C18 within MCPCM. Specifically, C18 experiences a homogeneous nucleation mode during solidification when not encapsulated, while C18 within the microcapsules undergoes both homogeneous and heterogeneous nucleation modes during solidification [44]. Figure 7d shows that the onset solidification temperature of MCPCM is higher than that of C18. This could be attributed to the presence of SG, which not only increases the thermal conductivity of MCPCM but also promotes the heterogeneous nucleation of C18 within MCPCM during solidification, resulting in an elevated solidification temperature [45,46].

The characterizations of the thermal performance of C18 and the different MCPCM are summarized accordingly in Table 3. From Table 3, we can see that *E_r_* increase with the SG content for MCPCM-1SG, MCPCM-2SG, and MCPCM-3SG. This is because during the preparation of MCPCM, the amphiphilic structure of SG allows it to be better distributed on the surface of C18 droplets to form an SG protective layer. After that, a MUF shell is formed on the SG layer by in situ polymerization. Therefore, when the SG content is suitable, the mixed shell composed of SG and MUF has a positive protective effect on C18, thus improving the energy storage capacity of the microcapsules.

### 3.5. Water Contact Angle of MCPCM

Permeability resistance is a key aspect of microencapsulation for practical applications. The core material of microcapsules with low impermeability migrates to the surface of the microcapsules over time, resulting in enhanced hydrophobicity of the microcapsules. Therefore, the water contact angle of microcapsules can be used as a parameter to evaluate the impermeability of MCPCM. To investigate the hydrophobic properties of MCPCM with varying SG contents and core/shell ratios, we conducted water contact angle tests (refer to Figure 8).

As shown in Figure 8a, MCPCM-0SG has the highest contact angle, while MCPCM-3SG has the lowest, suggesting that the introduction of SG has a significant impact on the hydrophobicity of MCPCM. As indicated in Table 3, the encapsulation rate of C18 in MCPCM-0SG was 90.22%, and the shell content was only 9.78%. This suggests that the shell layer of the microcapsules was thin, allowing low surface energy C18 molecules to migrate more easily from the interior of the microcapsules and be adsorbed on the surface of the microcapsules, thereby increasing the hydrophobicity of the MCPCM. With increasing SG content, the water contact angle of MCPCM decreases. The contact angle of MCPCM-3SG is 116°, which is 26° lower than that of MCPCM-0SG. This difference may be due to the higher anti-permeability of SG, which enhances the anti-permeability of the microcapsule shell, ultimately effectively preventing the migration of C18 molecules to the surface of the microcapsules and reducing the hydrophobicity of the microcapsules. As seen in Figure 8b, the water contact angle of MCPCM increases as the core–shell ratio increases, with MCPCM-4/1 exhibiting the highest contact angle (134°).

### 3.6. Leakage Prevention of MCPCM

To assess the leakage resistance of the microcapsules at elevated temperatures, the leakage rate was used for evaluating the leakage prevention performance of MCPCM. Figure 9 illustrates the leakage rates of the core materials of MCPCM. As shown in Figure 9a, the leakage rate of MCPCM with different SG contents was measured. The results indicate that the addition of SG led to a substantial reduction in the leakage rate of MCPCM. The core materials leakage rate and decreasing ratio of the leakage rate of MCPCM with different SG contents are listed in Table 4. Table 4 shows the leakage rates of MCPCM-0SG, MCPCM-1SG, MCPCM-2SG, and MCPCM-3SG within 8 h, which were 8.3%, 6.8%, 2.5%, and 1.6%, respectively. The leakage rates decreased significantly with increasing SG content. Compared to MCPCM-0SG, the leakage rates of MCPCM-2SG and MCPCM-3SG reduced by 69.9% and 80.7%, respectively, indicating a substantial decrease in the core leakage rate of the microcapsules. The introduction of SG as a stabilizer for Pickering emulsions creates a protective layer on the surface of the emulsion droplets, stabilizing the phase change material emulsion. The MUF polymer shell layer is then polymerized on the surface of the emulsion droplets, producing microcapsules with a hybrid MUF/SG shell layer that encapsulates the phase change material. Therefore, the appropriate amount of SG effectively reduces the leakage rate of the MCPCM core material.

Figure 9b depicts the leakage rates of MCPCM with different core/shell ratios, and the corresponding data can be found in Table 5. The results demonstrate that the leakage rate of MCPCM increases progressively with an increase in the core/shell ratio. This is because a thinner shell layer is formed as the core content increases and the MUF prepolymer content decreases. Therefore, the capability of the shell layer to prevent core structure leakage is reduced. Table 5 shows that the leakage rates of MCPCM-1/1, MCPCM-2/1, MCPCM-3/1, and MCPCM-4/1 were 1.8%, 2.5%, 4.3%, and 16.2%, respectively, within 8 h. Notably, MCPCM-4/1 exhibited a significant increase in leakage rate. This can be attributed to the fact that the MUF polymer shell layer is unable to effectively coat C18 when the core/shell ratio is relatively large, even in the presence of highly impermeable SG.
$$Dr\left(\%\right)=\frac{{Lr}_{MCPCM-0SG}-{Lr}_{MCPCM-nSG}}{{Lr}_{MCPCM-0SG}}\times 100\%.$$

### 3.7. Mechanical Strength of MCPCM

Mechanical strength is an essential requirement for the practical application of microcapsules. It is widely recognized that microcapsule shells with low mechanical strength may rupture under high-speed centrifugation due to intense collisions. Therefore, the breakage rate under high-speed centrifugation can serve as a measure of microcapsule mechanical strength. The breakage rates of MCPCM are presented in Figure 10. As observed from Figure 10a, the breakage rate of MCPCM-0SG was 3.3%, which was significantly higher than the breakage rates of MCPCM-1SG, MCPCM-2SG, and MCPCM-3SG upon the introduction of SG. The decrease in the breakage rate of MCPCM after introducing SG indicates that SG can effectively improve the mechanical strength of MCPCM. This is because SG acts as a Pickering particle stabilizer, forming a protective layer on the surface of the C18 droplet, and the prepolymer of MUF polymerizes on the surface of the SG-stabilized C18 emulsion droplet. The resulting MCPCM has a structure consisting of SG and MUF polymers. Such a hybrid MUF/SG shell layer structure can effectively enhance the mechanical strength of the microcapsules and reduce the breakage rate.

The breakage rates of MCPCM with different core/shell ratios are presented in Figure 10b, where MCPCM-1/1, MCPCM-2/1, MCPCM-3/1, and MCPCM-4/1 have breakage rates of 1.1%, 1.3%, 1.5%, and 3.3%, respectively. The results indicate that increasing the core/shell ratio results in a decrease in the mechanical strength of MCPCM. Moreover, microcapsule shell layers with too large core/shell ratios often have defects such as holes and fissures, which explains the higher breakage rate of MCPCM-4/1.

## 4. Conclusions

In this study, a novel MCPCM with MUF/SG hybrid shells was prepared by in situ polymerization using SG-stabilized Pickering emulsion as a template. Meanwhile, this study investigated the effects of different SG contents and core–shell ratios on the morphology, thermal properties, leak-proof properties, and mechanical strength of MCPCM. The results showed that the contact angle of MCPCM-3SG was measured as 116°, indicating a 26° reduction compared to MCPCM without SG. The leakage rate of MCPCM-3SG was measured as 1.6%, which is 80.7% lower than that of MCPCM without SG. Moreover, the breakage rate after high-speed centrifugation was measured as 1.2%, indicating a 63.6% reduction for MCPCM-3SG compared to MCPCM without SG. These findings suggest that the incorporation of SG can effectively improve the mechanical strength and leak-proof performance of MCPCM. Therefore, due to their outstanding characteristics, including high mechanical strength and excellent leak-proof performance, the obtained MCPCM have significant potential for applications in thermal energy management.

## Figures and Tables

**Figure 1 polymers-15-02441-f001:**
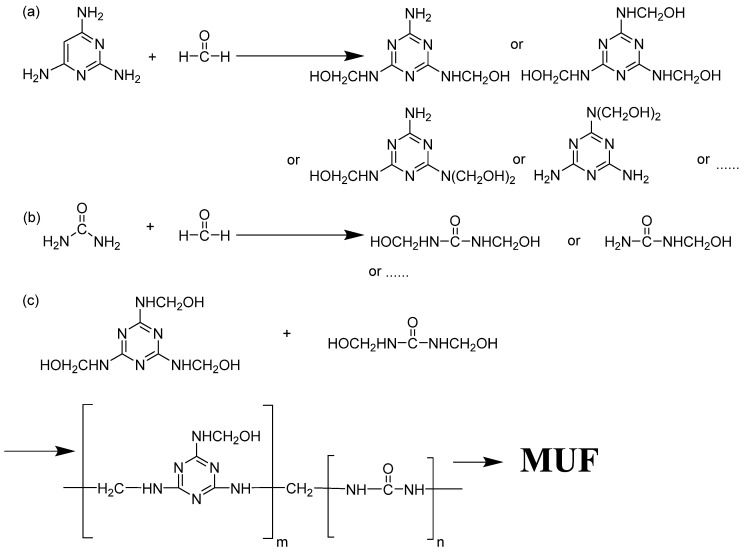
Reaction scheme for the formation of MUF shell during microencapsulation: (**a**) Nucleophilic addition reaction of melamine and formaldehyde under alkaline heating conditions. (**b**) Nucleophilic addition reaction of urea and formaldehyde under alkaline heating conditions. (**c**) Hydroxymethyl melamine and hydroxymethyl urea react under acidic conditions.

**Figure 2 polymers-15-02441-f002:**
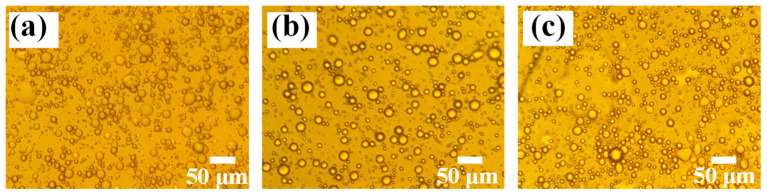
Polarized light micrographs of phase change material emulsions with different SG contents: (**a**) PCMEs-1SG, (**b**) PCMEs-2SG, and (**c**) PCMEs-3SG.

**Figure 3 polymers-15-02441-f003:**
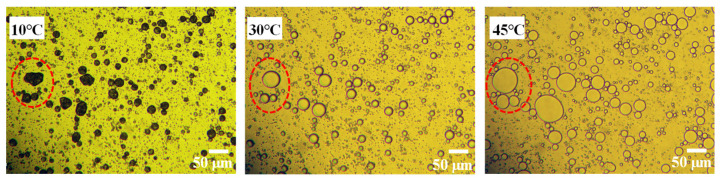
Polarized photomicrographs of PCMEs-2SG at different temperatures.

**Figure 4 polymers-15-02441-f004:**
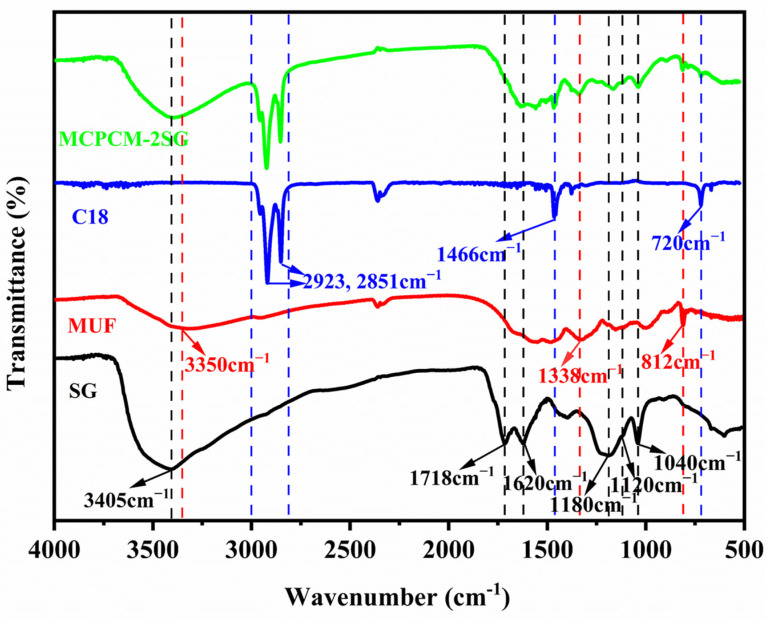
FT-IR spectra of MCPCM-2SG, C18, MUF resin, and SG.

**Figure 5 polymers-15-02441-f005:**
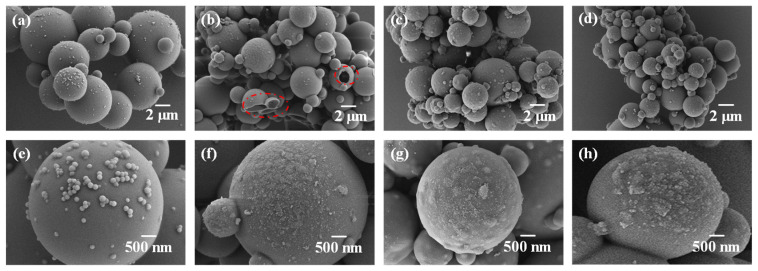
Scanning electron microscopy of MCPCM different amounts of SG added: (**a**,**e**) MCPCM-0SG; (**b**,**f**) MCPCM-1SG; (**c**,**g**) MCPCM-2SG and (**d**,**h**) MCPCM-3SG.

**Figure 6 polymers-15-02441-f006:**
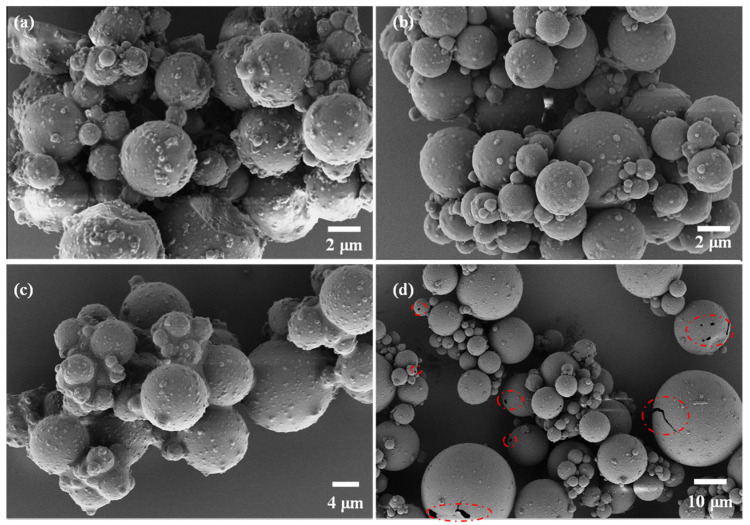
Scanning electron microscopy of MCPCM with different core/shell ratios: (**a**) MCPCM-1/1; (**b**) MCPCM-2/1; (**c**) MCPCM-3/1 and (**d**) MCPCM-4/1.

**Figure 7 polymers-15-02441-f007:**
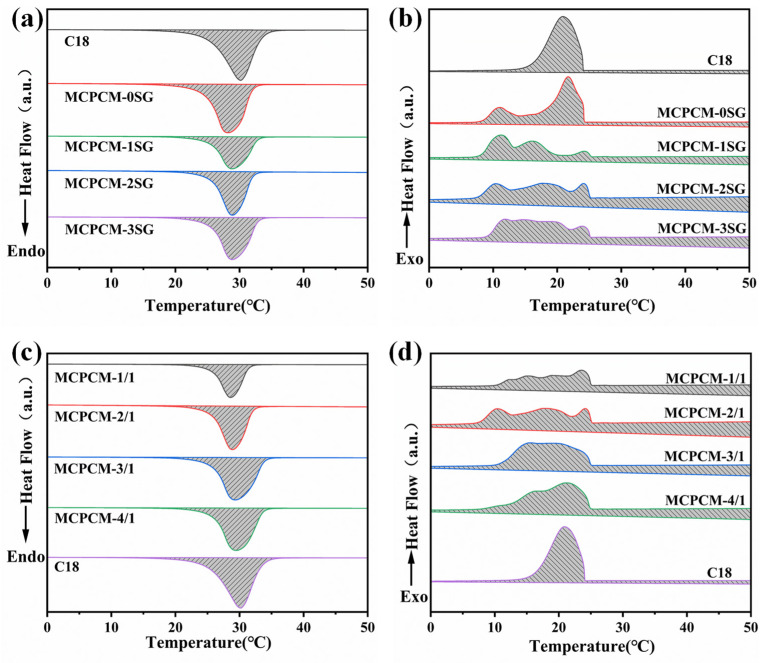
(**a**) DSC melting curves of C18 and MCPCM with different SG contents; (**b**) DSC solidification curves of C18 and MCPCM with different SG contents; (**c**) DSC melting curves of C18 and MCPCM with different core/shell ratio; (**d**) DSC solidification curves of C18 and MCPCM with different core/shell ratio.

**Figure 8 polymers-15-02441-f008:**
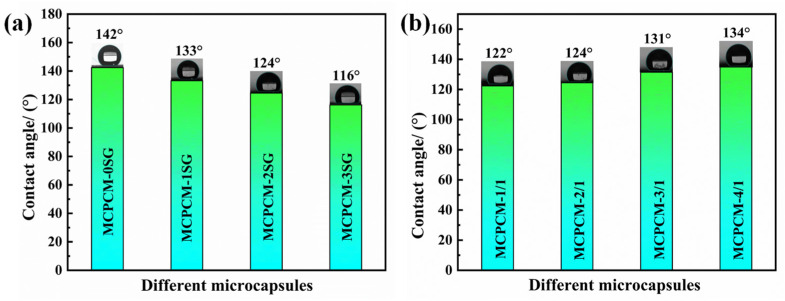
(**a**) Water contact angle of MCPCM with different SG contents. (**b**) Water contact angle of MCPCM with different core/shell ratios.

**Figure 9 polymers-15-02441-f009:**
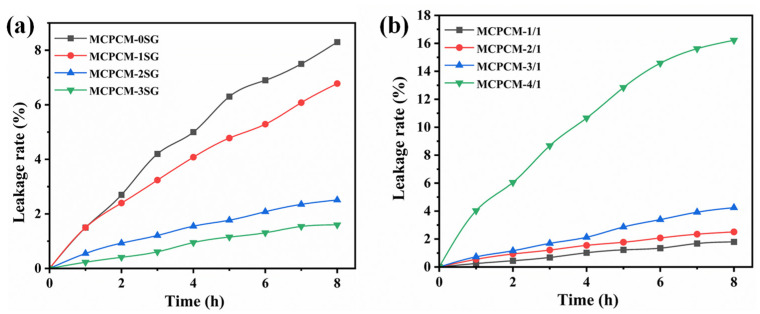
(**a**) The core materials leakage rate of MCPCM with different SG contents. (**b**) The core materials leakage rate of MCPCM with different core/shell ratios.

**Figure 10 polymers-15-02441-f010:**
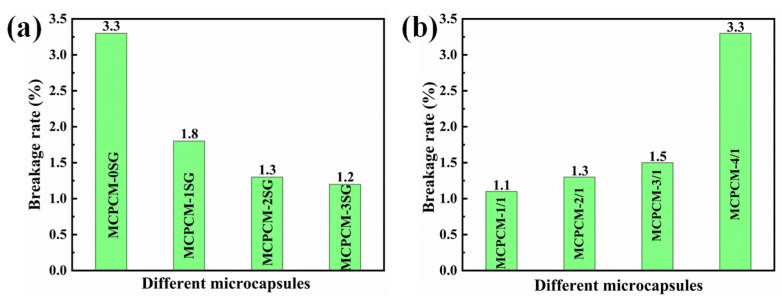
(**a**) Mechanical strength of microcapsules with different SG contents. (**b**) Mechanical strength of microcapsules with different core/shell ratios.

**Table 1 polymers-15-02441-t001:** Formulations for the preparation of PCMEs with different SG contents.

Sample	C18 (g)	SG Slurry (g)	SDS (g)	Deionized Water (mL)
PCMEs-0SG	15.00	0.00	1.50	90
PCMEs-1SG	15.00	1.50	0.00	90
PCMEs-2SG	15.00	3.00	0.00	90
PCMEs-3SG	15.00	4.50	0.00	90

**Table 2 polymers-15-02441-t002:** Formulations for the preparation of MCPCM with different SG contents and different core/shell ratios.

Sample	C18 (g)	SG Slurry (g)	Melamine (g)	Formaldehyde (g)	Urea (g)	Core/Shell (Mass Ratio)
MCPCM-0SG	15.00	0.00	3.81	6.89	0.92	2:1
MCPCM-1SG	15.00	1.50	3.81	6.89	0.92	2:1
MCPCM-2SG	15.00	3.00	3.81	6.89	0.92	2:1
MCPCM-3SG	15.00	4.50	3.81	6.89	0.92	2:1
MCPCM-1/1	7.50	1.50	3.81	6.89	0.92	1:1
MCPCM-2/1	15.00	3.00	3.81	6.89	0.92	2:1
MCPCM-3/1	22.50	4.50	3.81	6.89	0.92	3:1
MCPCM-4/1	30.00	6.00	3.81	6.89	0.92	4:1

**Table 3 polymers-15-02441-t003:** Thermal properties of C18 and different MCPCM.

Samples	*T_m_*/Peak (°C)	∆*H_m_* (J/g)	*T_c_*/Peak (°C)	∆*H_c_* (J/g)	*E_r_* (%)
C18	30.14	196.04	20.87	197.26	-
MCPCM-0SG	28.19	177.48	21.71	177.36	90.22
MCPCM-1SG	28.83	119.36	11.21	119.96	60.85
MCPCM-2SG	28.88	140.35	17.63	139.70	71.21
MCPCM-3SG	28.83	157.32	11.85	157.89	80.14
MCPCM-1/1	28.60	93.20	23.53	93.31	47.42
MCPCM-2/1	28.88	140.35	17.63	139.70	71.21
MCPCM-3/1	29.31	174.27	15.14	174.12	88.58
MCPCM-4/1	29.47	165.52	21.10	167.23	84.60

**Table 4 polymers-15-02441-t004:** The core materials leakage rate of MCPCM with different SG contents according to time.

Time (h)	MCPCM-0SG	MCPCM-1SG		MCPCM-2SG		MCPCM-3SG	
	*L_r_* (%)	*L_r_* (%)	*Dr*_1_ ^a^ (%)	*L_r_* (%)	*Dr*_2_ ^a^ (%)	*L_r_* (%)	*Dr*_3_ ^a^ (%)
2	2.7	2.4	11.1	0.9	66.7	0.41	85.2
4	5.0	4.1	18.0	1.6	68.0	0.95	80.0
6	6.9	5.3	23.2	2.1	69.6	1.31	81.2
8	8.3	6.8	18.1	2.5	69.9	1.62	80.7

^a^ The decreasing ratio of the leakage rate.

**Table 5 polymers-15-02441-t005:** The core materials leakage rate of MCPCM with different core/shell ratios.

Time (h)	MCPCM-1/1 *L_r_* (%)	MCPCM-2/1 *L_r_* (%)	MCPCM-3/1 *L_r_* (%)	MCPCM-4/1 *L_r_* (%)
2	0.5	0.9	1.2	6.1
4	1.0	1.6	2.1	10.7
6	1.4	2.1	3.4	14.6
8	1.8	2.5	4.3	16.2

## Data Availability

Data are contained within the article.

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
