# Peer review of "Preparation of Microencapsulated Phase Change Materials from Sulfonated Graphene Stabilized Pickering Emulsion"

_polymers, 2023, doi:10.3390/polym15112441_

Round 1

Reviewer 1 Report

The manuscript under consideration is devoted to the development of a method for obtaining microcapsulated paraffin C18 for use as a fusible heat-preserving substance. The effect of heat saving is as follows. When cooling the liquid hot paraffin obtained by heating it above the melting point, a phase transition occurs - its crystallization begins. Since crystallization is an exothermic process, the temperature of the material remains constant during the crystallization process, which is perceived as heat preservation . However, the use of liquid paraffin is inconvenient due to the possibility of its leakage. So, it is very attractive to use a crystallizable agent not in the form of a liquid, but in the form of a polymer powder. In this regard, the task in this paper is to obtain crystallizable agent  in the form of a solid powder, the particles of which have a "core /shell" structure, while paraffin is in the core, and the shell material is a polymer obtained by the reaction of melamine and urea with formaldehyde.

The article describes the operations for the preparation of the target product through the intermediate stage of obtaining paraffin emulsion in water by the Pickering method – without the use of surfactants - using microparticles of sulfonated graphite in the form of a solid stabilizing agent. Separately, the preparation of a melamine-urea-formaldehyde prepolymer in the form of a solution in water is carried out. After merging the two solutions and acidifying the system to pH = 5, polycondensation is carried out for 20-30 minutes, after which the reaction is stopped by bringing the pH to 8.5, and the microcapsulated product is isolated. The structure was characterized by IR spectroscopy and polarization optical microscopy. The functional properties of the microcapsulated product were also determined: heat release during crystallization by the DSC method. The mechanical properties of the shell were evaluated by conducting tests on a centrifuge in the developed mode. The degree of paraffin leakage after holding microcapsules at elevated temperature was also evaluated. As a result of a systematic series of experiments, optimal ratios of the initial components (paraffin-sulfonated graphite- melamine-formaldehyde-urea) and optimal modes of operations of the initial formulations were determined, which allowed to obtain the best final results.

As comments, the following can be noted: the manuscript does not characterize the phase state of the polymerization system in the process of obtaining microcapsules. The comment is need explaining how the polymer shell is formed on the emulsion droplets? What is the presumed topology of the shell macromolecules, what is their phase-morphological state?

Manuscript has to be supplemented with necessary information before it will be recommended for publication.

Minor editing is reqired

Author Response

Response 1: In the revised manuscript, we have included a detailed explanation of how the polymer shell is formed on the emulsion droplets and speculating on the polymer macromolecules that may be formed. The corresponding findings are depicted in Figure 1.

Changes: p4 143-154

Response 2: The prepolymer preparation stage of polymer shells, where both raw materials and products are in liquid form. The synthesis stage of the polymer shell, with the polymerization reaction, the molecular weight of the product increases forming the cross-linked structure of the water-insoluble MUF polymer; this process reacts with the raw material in the liquid state and the product in the solid state.

Reviewer 2 Report

This paper prepared a novel MCPCM with hybrid shells of mela-mine-urea-formaldehyde (MUF) and sulfonated graphene (SG) by in-situ polymeri-zation using SG-stabilized Pickering emulsion as a template and analyzing the effects of SG content and core/shell ratio on the morphology, thermal properties, leak-proof properties, and mechanical strength of the MCPCM. 

This paper is well-written and gives helpful information to readers. However, some parts need to be modified before publication. Other comments:

1. The article listed the materials currently used to prepare MCPCM, but lacked an examination of the limitations of previous studies. To highlight the excellence of the materials used in this study. 2. Why did the Pickering emulsion of sulfonated graphene prepare MCPCM? What other authors or research have already performed similar analyses? 3. References are required for the description of the chemical composition of MCPCM in section 3.2. References need to be added for the chemical composition corresponding to each absorption spectrum. 4. The thermal properties of the different MCPCMs are presented in Table 3, but the results for the MCPCM-2/1 sample are missing. 5. The conclusion should specifically state the findings of this study rather than repeating the experimental results.

The quality of language is good 

Author Response

We appreciate the reviewer's feedback and have addressed the following concerns:

Comment 1: The article listed the materials currently used to prepare MCPCM, but lacked an examination of the limitations of previous studies. To highlight the excellence of the materials used in this study.

Response 1: We have added a section to the introduction discussing the limitations of previous studies in preparing MCPCM and how our study addresses these limitations. We also highlighted the advantages of the materials used in our study.

Changes: p1 44-61

Comment 2: Why did the Pickering emulsion of sulfonated graphene prepare MCPCM? What other authors or research have already performed similar analyses?

Response 2: We have added a paragraph in the introduction explaining why we used sulfonated graphene-stabilized Pickering emulsion to prepare MCPCM.

Changes: p2 64-85

Comment 3: References are required for the description of the chemical composition of MCPCM in section 3.2. References need to be added for the chemical composition corresponding to each absorption spectrum.

Response 3: We have added references to the chemical composition of MCPCM in section 3.2 and for each absorption spectrum presented.

Changes: Added references [12][41]and [42].

Comment 4: The thermal properties of the different MCPCMs are presented in Table 3, but the results for the MCPCM-2/1 sample are missing.

Response 4: We apologize for the oversight in Table 3 and have included the thermal properties for the MCPCM-2/1 sample.

Changes: p9 Table 3.

Comment 5: The conclusion should specifically state the findings of this study rather than repeating the experimental results.

Response 5: We revised the conclusion to more specifically state the findings of our study rather than repeating the experimental results.

Changes: p12 387-389